# Highly efficient and air-stable Eu(II)-containing azacryptates ready for organic light-emitting diodes

Jiayi Li[1,2], Liding Wang[1,2], Zifeng Zhao [1,2], Boxun Sun[1], Ge Zhan[1], Huanyu Liu[1], Zuqiang Bian [1] & Zhiwei Liu [1✉]

Divalent europium 5d-4f transition has aroused great attention in many fields, in a way of doping $Eu^{2+}$ ions into inorganic solids. However, molecular $Eu^{2+}$ complexes with 5d-4f transition are thought to be too air-unstable to explore their applications. In this work, we synthesized four $Eu^{2+}$-containing azacryptates $EuX_2$-$N_n$ ($X = Br$, I, $n = 4$, 8) and systematically studied the photophysical properties in crystalline samples and solutions. Intriguingly, the $EuX_2$-$N_8$ complexes exhibit near-unity photoluminescence quantum yield, good air-/thermal-stability and mechanochromic property ($X = I$). Furthermore, we proved the application of $Eu^{2+}$ complexes in organic light-emitting diodes (OLEDs) with high efficiency and luminance. The optimized device employing $EuI_2$-$N_8$ as emitter has the best performance as the maximum luminance, current efficiency, and external quantum efficiency up to $25470\ cd\ m^{-2}$, $62.4\ cd\ A^{-1}$, and 17.7%, respectively. Our work deepens the understanding of structure-property relationship in molecular $Eu^{2+}$ complexes and could inspire further research on application in OLEDs.

[1] Beijing National Laboratory for Molecular Sciences (BNLMS), State Key Laboratory of Rare Earth Materials Chemistry and Applications, Beijing Engineering Technology Research Centre of Active Display, College of Chemistry and Molecular Engineering, Peking University, 100871 Beijing, China. [2] These authors contributed equally: Jiayi Li, Liding Wang, Zifeng Zhao. ✉email: zwliu@pku.edu.cn

The 5d–4f transition in lanthanides (Ln) has been studied for decades on the luminescent mechanism and potential applications in various fields[1–8]. For the well-established luminescence of lanthanide ions, f–f transition can be observed under ambient conditions, while the 5d–4f transition is usually absent due to thermally quenching by fast intersystem crossing from $4f^{n-1}5d^1$ to $4f^n$ configuration. In divalent lanthanide systems, 5d–4f transition is much more prominent for its spin-allowed nature and the stabilization of the 5d orbitals[6–8].

Among all $Ln^{2+}$ ions, $Eu^{2+}$ ions exhibit strong 5d–4f transition and great applications for two reasons: (1) the 5d level is near or below $^6P_{7/2}$, decreasing the multiphoton relaxation[6], (2) the reduction potential of $Eu^{3+}/Eu^{2+}$ is not too negative. The research on luminescent properties of $Eu^{2+}$ ions can be roughly divided into two categories: $Eu^{2+}$ dopants in inorganic matrix and molecular $Eu^{2+}$ complexes. The first one has been extensively studied while the latter remains unexplored in many aspects. The physicochemical properties of $Eu^{2+}$ complexes are mainly studied using cyclopentadienyls, hydrotris(pyrazolyl)borates, silylamides and their derivatives as ligands[9–11]. Recently, Allen et al. reported series of $Eu^{2+}$-containing azacryptate complexes which have aroused growing interest for their attractive luminescent properties, photoredox catalytic performance and magnetic resonance imaging[3,12–15].

The uniqueness of 5d–4f luminescent mechanism enables $Eu^{2+}$ complexes to have great potential in high-performance organic light-emitting diodes (OLEDs), a technology has successfully been commercialized in cutting-edge displays and is under developing in solid-state lighting. To reach 100% theoretical exciton utilization efficiency (EUE), which is the key parameter to enhance the energy efficiency, phosphorescence[16,17], thermally activated delayed fluorescence (TADF)[18,19] and organic radical materials[20] were discovered in succession and applied as emitters in OLEDs. Comparing with the traditional f–f transition and other currently used emitters, divalent europium compounds have the following significant advantages: (i) short decay lifetime: the f–f transition is spin-forbidden with long lifetimes up to milliseconds, strongly limiting their maximum luminance, while 5d–4f transition is spin-allowed with typical lifetimes in nanosecond scale, which significantly reduce the excited-state quenching to reach higher luminance and lower efficiency roll-off, (ii) high EUE: $Eu^{2+}$ ion exhibits a unique transition for the open-shell electron from $4f^65d^1$ to $4f^7$, which can harvest 100% exciton energy theoretically[20,21], (iii) easily tunable emission by varying coordinate environment: the 5d orbitals are sensitive to the ligand field while the 4f orbitals, effectively shielded by 5s5p, are not sensitive to surroundings[22–25]. (iv) high abundance: europium has crustal abundance of $10^{-6}$ wt, much higher than the noble metals (Ir, Pt) used in commercial OLEDs. Thus, we believe that 5d–4f transition materials, represented by $Eu^{2+}$ complexes, will be the next unexplored but promising field in OLED emitters.

Despite the advantages mentioned, $Eu^{2+}$ complexes are strongly limited by their air stability according to the standard potential $\varphi(Eu^{3+}/Eu^{2+}) = -0.38$ V. To the best of our knowledge, there was only one report of OLED device based on $Eu^{2+}$ complexes, with unsatisfied performance in external quantum efficiency (EQE) of 0.01% and maximum luminance of 10 cd m$^{-2}$ considering the high photoluminescence quantum yield (PLQY) of the complex to be 85%[26]. Thus, more efforts must be worked on the rational design of $Eu^{2+}$ complexes and the deep understanding of the electroluminescent process to boost efficiency and luminance. We propose that the steric effect of cryptate ligands and coordinate interaction could improve the stability of $Eu^{2+}$ complexes. The steric effect prevents $Eu^{2+}$ from $O_2$ by a more rigid structure. Improving the coordination interaction between the ligand and $Eu^{2+}$ can largely enhance the thermodynamic stability. Thus, two ligands, 1,4,7,10-tetraazacyclododecane ($N_4$)

and 1,4,7,10,13,16,21,24-octaazabicyclo[8.8.8]hexacosane ($N_8$) are chosen for the design of four $Eu^{2+}$-containing azacryptates named as $EuX_2–N_n$ ($X = Br$, I, $n = 4$, 8). Series of crystal analysis, spectral, stability, and theoretical studies were undertaken to reveal the photophysical nature of these $Eu^{2+}$ complexes. Then $EuX_2–N_8$ complexes were chosen as emitters in OLEDs for their high efficiency and good thermal/air stability. As a breakthrough, the optimized device using $EuI_2–N_8$ exhibits excellent performance with a maximum EQE of 17.7% and a maximum luminance of 25470 cd m$^{-2}$.

## Results

**Synthesis and structural analysis**. The four $Eu^{2+}$ complexes $EuX_2–N_n$ ($X = Br$, I, $n = 4$, 8) were synthesized in glovebox by mixing $EuX_2$ and corresponding ligands in methanol[3,27]. The purified products were identified by elemental analysis. Then, single-crystal X-ray diffraction (SCXRD) was performed to investigate the coordinate geometry of these $Eu^{2+}$-containing azacryptates (Fig. 1). $EuBr_2–N_4$ crystallizes in space group P21/n and one unit contains two azacryptate cations, four bromide ions in the outer sphere and four methanol. The $N_4$ ligands have two possible conformations, 50% for each. Thus, the $Eu^{2+}$ center, coordinated by eight nitrogen atoms from two ligands, adopts an unusual geometry with averagely half in square antiprism and half in distorted cube. Likely, the same coordinate geometry is found in $EuI_2–N_4$, which crystallizes in a higher-symmetry space group of Cmca without solvent. There are two sets of $[Eu(N_4)_2]^{2+}$ with different orientations in one cell, locating in the edge center and body center, while the eight iodide ions intersperse therein (see Supplementary Fig. 1). The crystal structures of $EuX_2–N_8$ show that the center $Eu^{2+}$ is coordinated by eight nitrogen atoms and one halide ion as a distorted "hula-hoop" geometry, with the other halide in the outer sphere as a counterion, which are consistent with the reported structures[1,3]. As shown in Table 1, the bond lengths of Eu-N in $EuX_2–N_4$ are relatively shorter than those in $EuX_2–N_8$, indicating the $N_4$ complexes have stronger coordinate interaction between $Eu^{2+}$ and ligands. Considering the charge separation in crystals, the $EuX_2–N_4$ compounds behave more like ionic crystals with relatively stronger electrostatic attraction between the counterion halides and the $[Eu(N_4)_2]^{2+}$ ions.

**Photophysical properties**. To systematically study the photophysical properties of $Eu^{2+}$ complexes, steady-state spectra and transient spectra were collected. Crystalline powder of $EuX_2–N_4$ shows orange-red emissions with maximum wavelength ($\lambda_{max}$) of 605 nm ($X = Br$) and 613 nm ($X = I$), respectively (Fig. 2). Changing the azacryptates from $N_4$ to $N_8$, the $EuX_2–N_8$ complexes exhibit strong hypsochromic shift induced by the weaker crystal field of $N_8$ ligands, with $\lambda_{max}$ of 510 nm ($X = Br$) and 515 nm ($X = I$). The lifetimes for these complexes were found to be hundreds of nanoseconds (Table 2, Supplementary Fig. 2), within the expected range for 5d–4f transition[3,28]. Full widths at half maximums (FWHMs) for these complexes in solid powder (40–45 nm) are relatively narrow comparing with luminescent materials featuring in charge-transfer (CT) mechanisms. The excitation bands of these complexes are broad and featureless, ranging from 230 to 500 nm ($EuX_2–N_8$) and 230 to 600 nm ($EuX_2–N_4$) as shown in Supplementary Fig. 3. Based on aforementioned photophysical studies and considering that the ligands in our system are saturated organic compounds with high-energy levels, it is reasonable to rule out the possibilities of ligand-metal charge transfer (LMCT). Hence, the excitation and emission processes can be regarded as the electronic transitions in $Eu^{2+}$ ion, where the ground state is $4f^7$ [$^8S_{7/2}$] and the excitation state is $4f^6[^7F_0]5d^1$, specifically.

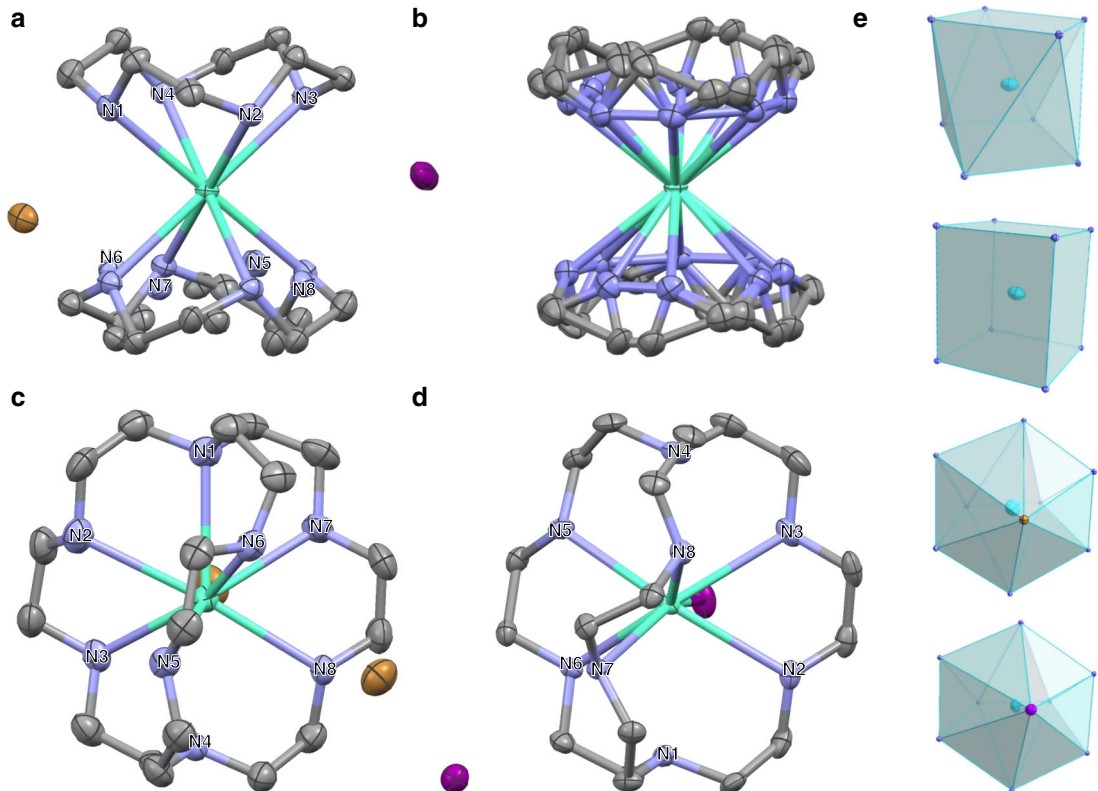

**Fig. 1 The crystal structure of four Eu$^{2+}$ complexes.** ORTEP drawings of the crystal structures of (**a**) EuBr$_2$–N$_4$, (**b**) EuI$_2$–N$_4$, (**c**) EuBr$_2$–N$_8$, and (**d**) EuI$_2$–N$_8$, respectively. **e** The coordination polyhedrons (from top to bottom): square antiprism in EuX$_2$–N$_4$, distorted cube in EuX$_2$–N$_4$, distorted "hula-hoop" in EuBr$_2$–N$_8$ and EuI$_2$–N$_8$. The nitrogen atoms are labeled from N1–N8 for the data in Table 1. The solvent methanol (in EuBr$_2$–N$_4$ and EuI$_2$–N$_8$) and all the hydrogens are omitted for clarification. Atom notation: Eu (cyan), C (gray), N (blue), Br (brown), I (purple).

### Table 1 The bond lengths (distances) around Eu$^{2+}$ center in EuX$_2$-N$_n$.

|  | EuBr$_2$-N$_4$ | EuI$_2$-N$_4$ | EuBr$_2$-N$_8$ | EuI$_2$-N$_8$ |
|---|---|---|---|---|
| Eu–X[a] | 4.9097(3) | 5.3019(2) | 3.3129(8) | 3.6170(4) |
|  |  |  | 4.8539(11) | 4.8205(5) |
| Eu–N1[b] | 2.660(3) | 2.671(5) | 2.907(6) | 3.027(3) |
| Eu–N2 | 2.691(4) | 2.724(5) | 2.830(15) | 2.790(3) |
| Eu–N3 | 2.698(4) | 2.712(6) | 2.787(6) | 2.839(4) |
| Eu–N4 | 2.683(3) | 2.672(6) | 2.932(5) | 3.040(3) |
| Eu–N5 | 2.686(4) | 2.724(5) | 2.792(5) | 2.761(3) |
| Eu–N6 | 2.725(4) | 2.671(5) | 2.716(5) | 2.789(3) |
| Eu–N7 | 2.723(3) | 2.672(6) | 2.773(6) | 2.758(3) |
| Eu–N8 | 2.689(4) | 2.712(6) | 2.738(12) | 2.819(3) |

[a]The EuX$_2$-N$_4$ structures only have one Eu–X distance, while in EuX$_2$-N$_8$, one halide directly bonds to center Eu$^{2+}$ ion (inner sphere) and the other is in the outer sphere.
[b]The labels for nitrogen atoms in EuX$_2$-N$_4$ is named as 1–4 and 5–8 for two N$_4$ ligands. The labels in EuX$_2$-N$_8$ are shown in Fig. 1.

Due to the insolubility of EuX$_2$–N$_4$ in common solvents, we only studied the photophysical properties of EuX$_2$–N$_8$ in methanol solution (1.5 mM) under N$_2$ atmosphere. The EuX$_2$–N$_8$ solutions show bright yellow emission with $\lambda_{max}$ of 579 nm, and the emission spectra of the two complexes are almost identical. The emission is red-shifted by about 70 nm comparing with their solid samples, which is presumed to be caused by the differences in conformation of N$_8$ ligand in solid and solution, outweighing the effect of different halogens. As shown in Fig. 2c, the excitation bands of two compounds are similarly located at 280 nm and 410 nm, attributed to the transition from $4f_{z3}$ to $5d_{z2}$ and from $4f_{z3}$ to $5d_{xy}$, respectively[29,30].

The UV–visible spectra (Supplementary Fig. 4) show that EuX$_2$–N$_8$ complexes have high-energy absorption around 250 nm ($\varepsilon > 1000$ L mol$^{-1}$ cm$^{-1}$) and low-energy absorption peak at 404 nm ($\varepsilon = 644$ L mol$^{-1}$ cm$^{-1}$, $X = $ Br) and 405 nm ($\varepsilon = 512$ L mol$^{-1}$ cm$^{-1}$, $X = $ I), respectively, which is consistent with their excitation bands. The large molar absorptivity is on par with the reported Eu$^{2+}$ complexes due to the Laporte- and spin-allowed nature of $f$–$d$ transition[10,11,31]. The time-dependent density functional theory (TD-DFT) calculation was conducted for EuX$_2$–N$_8$ and EuBr$_2$–N$_4$. The calculation prediction of EuX$_2$–N$_8$ is very close to the experimental data. For the N$_4$ complex, EuBr$_2$–N$_4$ has two possible conformations (high symmetry: cubic geometry and low symmetry: square antiprism), and calculation result suggests that the different conformations exhibit distinct absorption bands.

Interestingly, EuI$_2$–N$_8$ exhibits mechanochromic property, showing a fluorescence color change from green to yellow under moderate mechanical grinding in Fig. 2e. The emission spectra show that a new peak emerges in a longer wavelength region after grinding. Then the excitation and transient spectra of 515 nm and 580 nm in the ground sample were collected to probe possible explanation as shown in Supplementary Fig. 5 and Fig. 2f. The similar excitation characteristics and a slightly longer decay of the new peak at 580 nm infer that the longer-wavelength emission is still from Eu$^{2+}$ center at a marginally different coordination environment. We tentatively attribute the mechanochromic property to the change in ligand conformation upon grinding, resulted from the relatively weak lattice energy of EuI$_2$–N$_8$. And the reversible process is essentially recrystallization in certain solvent atmosphere, like methanol[32–34]. Furthermore, the longer-wavelength emission can be pronouncedly enhanced by fast precipitation in antisolvent

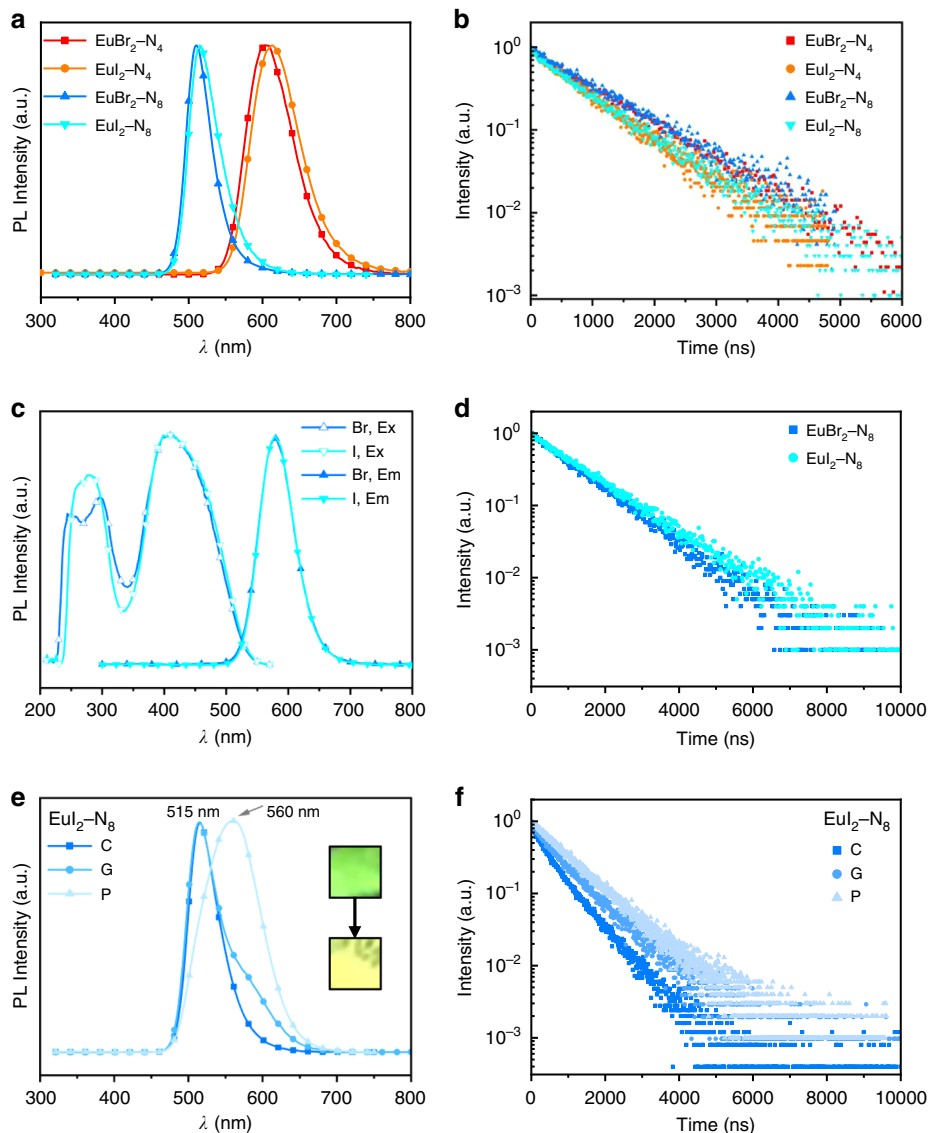

**Fig. 2 The photophysical properties of Eu$X_2$-N$_n$ compounds in solid and solution. a** The emission and (**b**) decay spectra of solid Eu$X_2$-N$_n$ ($X$ = Br, I, $n$ = 4, 8). **c** The excitation (Ex), emission (Em), and (**d**) decay spectra of Eu$X_2$-N$_8$ in methanol solution (1.5 mM). **e** The emission spectra of EuI$_2$-N$_8$ and (**f**) the decay spectra of EuI$_2$-N$_8$ of crystals (C), ground samples (G), and powder (P).

**Table 2 The summary of photoluminescent properties of the four Eu$X_2$-N$_n$ compounds.**

| State | EuBr$_2$-N$_4$ Solid | EuI$_2$-N$_4$ Solid | EuBr$_2$-N$_8$ Solid | EuBr$_2$-N$_8$ Solution | EuI$_2$-N$_8$ Solid | EuI$_2$-N$_8$ Solution |
|---|---|---|---|---|---|---|
| PLQY/% | 64 | 56 | ~100 | 55 | ~100 | 47 |
| FWHM/nm | 74 | 75 | 40 | 67 | 45 | 66 |
| $\lambda_{max}$/nm | 605 | 613 | 510 | 579 | 515 | 579 |
| Lifetime/ns | 1025 | 816 | 997 | 1179 | 864 | 1318 |

(tetrahydrofuran) to get amorphous powder of EuI$_2$–N$_8$. As shown in Fig. 2f, the emission shifts to a longer wavelength of 560 nm and the decay lifetime also increases from crystalline to amorphous state. However, a similar phenomenon was not observed in EuBr$_2$–N$_8$ (Supplementary Fig. 6), which indicates that EuBr$_2$–N$_8$ has a higher lattice energy, so it is harder to change the ligand conformation by such small mechanical stimulation.

**Thermal and air stability**. Thermal properties of these four compounds are studied by thermogravimetric analysis (TGA), which is of great significance for their further applications in OLEDs. The deposition temperature ($T_d$, corresponding to 5% weight loss) are around 270 °C, 265 °C, 393 °C, and 436 °C for EuBr$_2$–N$_4$, EuI$_2$–N$_4$, EuBr$_2$–N$_8$, and EuI$_2$–N$_8$ in Fig. 3a, respectively. After 550 °C, the unchanged residue weight percentages of these compounds should be the mass percentage of metal halides, for that the decomposition process is tentatively attributed to the break of coordinate bonds followed by sublimation of organic ligands. Then the relative error (RE) of residue weight is calculated to verify that speculation. As shown in Supplementary Table 2, the REs of Eu$X_2$–N$_4$ are reasonably low (~3%) while the REs of Eu$X_2$–N$_8$ are too high (~6% and ~10% for $X$ = Br, I, respectively). The element analysis is employed to exclude the possibility of impurities in Eu$X_2$–N$_8$. Thus, we believe these Eu$X_2$–N$_8$ compounds undergo both decomposition and sublimation around $T_d$, which will result in a large deviation in the final weight percentage.

Then the sublimation properties of these compounds were tested under high vacuum of $10^{-5}$ Pa and gradient heating. The

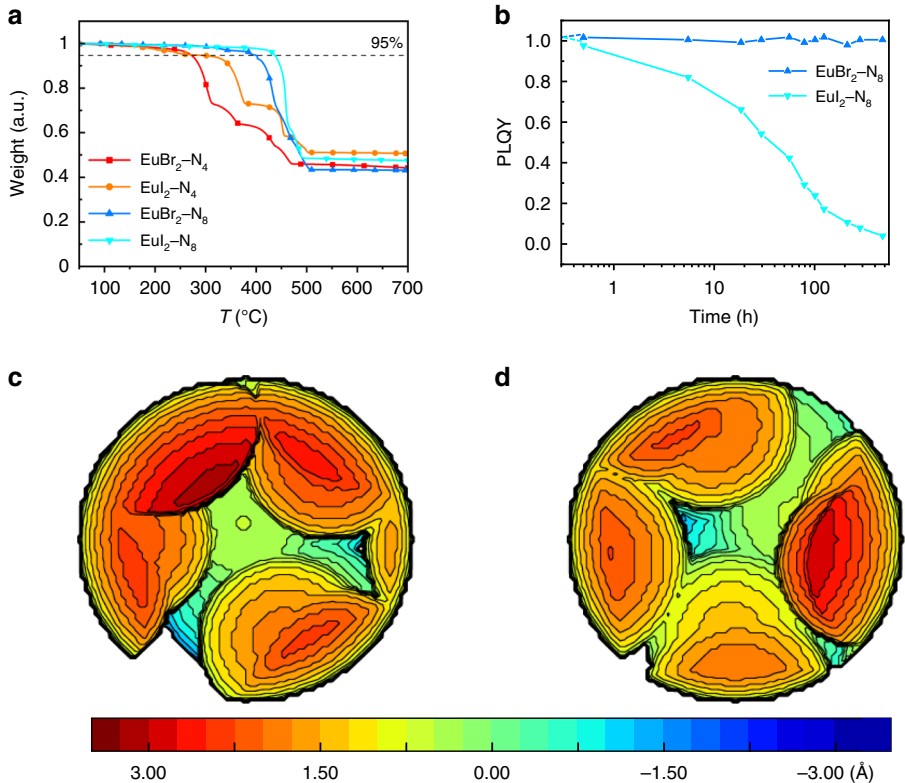

**Fig. 3 The thermal properties and buried volume calculation of EuX₂-Nₙ compounds. a** Thermogravimetric analysis (TGA) of $EuX_2$-$N_n$, where the 95%-weight line is shown to determine the $T_d$. **b** The change in PLQY of $EuX_2$-$N_8$ complexes in air. **c** Buried volume $\%V_{bur}$ calculated of $EuBr_2$-$N_8$ (77.0%). **d** Buried volume $\%V_{bur}$ calculated of $EuI_2$-$N_8$ (75.1%).

EuBr₂–N₈ was found to be completely sublimable around 320 °C (tube temperature, which is different from the sample temperature) at a small scale of 50 mg. It is notable that there will be obvious decomposition at large-scale sublimation, probably due to the uneven heating in the sublimation boat. A similar property is found for EuI₂–N₈ with a higher sublimation temperature at 350 °C.

Considering the high thermal stability and near-unity PLQY, the N₈ complexes are potential candidates used in OLEDs as emitters. However, Eu²⁺ ion is traditionally known to be easily oxidized to Eu³⁺ by O₂, hence the air stability is a critical parameter in terms of further applications. To shed light on their air stability at room temperature, the PLQYs of EuX₂–N₈ were measured as the function of time. As shown in Fig. 3b where the PLQY value change reflects their respective air stability, the quantum yield of EuBr₂–N₈ does not change after exposure in air over 450 h and EuI₂–N₈ is metastable towards air. To explain the differences in stability for future design of Eu²⁺ complexes, the analysis of buried volume ($\%V_{bur}$) was calculated to estimate the steric protection by N₈ ligands as shown in Fig. 3c, d[35,36]. $\%V_{bur}$ is defined as the fraction of volume of ligand over the total volume of sphere centered on the metal. It determines the steric effect of a given ligand regard to the first coordination sphere (Supplementary Fig. 7). The two complexes all exhibit high $\%V_{bur}$ values (77% to 75.1% for X = Br, I, respectively). EuBr₂–N₈ has a slightly higher $\%V_{bur}$ than EuI₂–Nₙ due to the closer distance between Eu²⁺ and Br⁻. Thus, the divergence in air stability between these two N₈ complexes is related to the different lattice energy considering the similar $\%V_{bur}$. The EuI₂–N₈ has smaller lattice energy than EuBr₂–N₈ because of weaker static interaction and the existing of solvent methanol in crystal, which is also applied to explain their different mechanochromic behaviors.

**Electroluminescence performance**. Based on the photophysical and stability studies, the EuX₂–N₈ complexes are better candidates used in OLEDs as emitters. Prior to device fabrication, the highest occupied molecular orbital (HOMO) and the lowest unoccupied molecular orbital (LUMO) energy levels of the two complexes were deduced from their ultraviolet photoelectron spectroscopy (Supplementary Fig. 8) and ultraviolet absorption spectra data. Then, great efforts have been devoted to optimizing the device structure due to the lack of experiences of Eu²⁺ complexes used in OLEDs. The EuBr₂–N₈ was first chosen for device optimization, which includes screening host materials, finding the best combination of hole transporting layer (HTL) and electron transporting layer (ETL), adjusting the thickness of the emission layer in Supplementary sections 1–4. Then, we followed the optimized conditions and further adjusted the doping concentration and the thickness of the emission layer of the EuI₂–N₈ device in Supplementary sections 5–6. The details of materials used, device optimization, and performance are shown in Supplementary Figs. 9–17 and Supplementary Tables 3–8.

On the base of the aforementioned process, the optimized OLED structure is ITO/MoO₃ (2 nm)/N,N'-bis(1-naphthalenyl)-N,N'-bis-phenyl-(1,1'-biphenyl)-4,4'-diamine (NPB, 50 nm)/cyclohexylidenebis[N,N'-bis(p-tolyl)aniline] (TAPC, 10 nm)/EuX₂–N₈:4,4',4''-tris[phenyl(m-tolyl)amino]triphenylamine (m-MTDATA, 25 nm)/diphenyl[4-(triphenylsilyl)phenyl]phosphine oxide (TSPO1, 10 nm)/4,7-diphenyl-1, 10-phenanthroline (Bphen, 30 nm)/LiF (0.7 nm)/Al. The best EuBr₂–N₈ device gives pretty good performance with a turn-on voltage ($V_{on}$) of 6.2 V, a maximum luminance ($L_{max}$) of 10,200 cd m⁻², a maximum current efficiency ($CE_{max}$) of 52.8 cd A⁻¹ and a maximum EQE of 15.5%. While the champion device is obtained by using EuI₂–N₈ as the emitter, the $V_{on}$, $L_{max}$, $CE_{max}$, and $EQE_{max}$ are

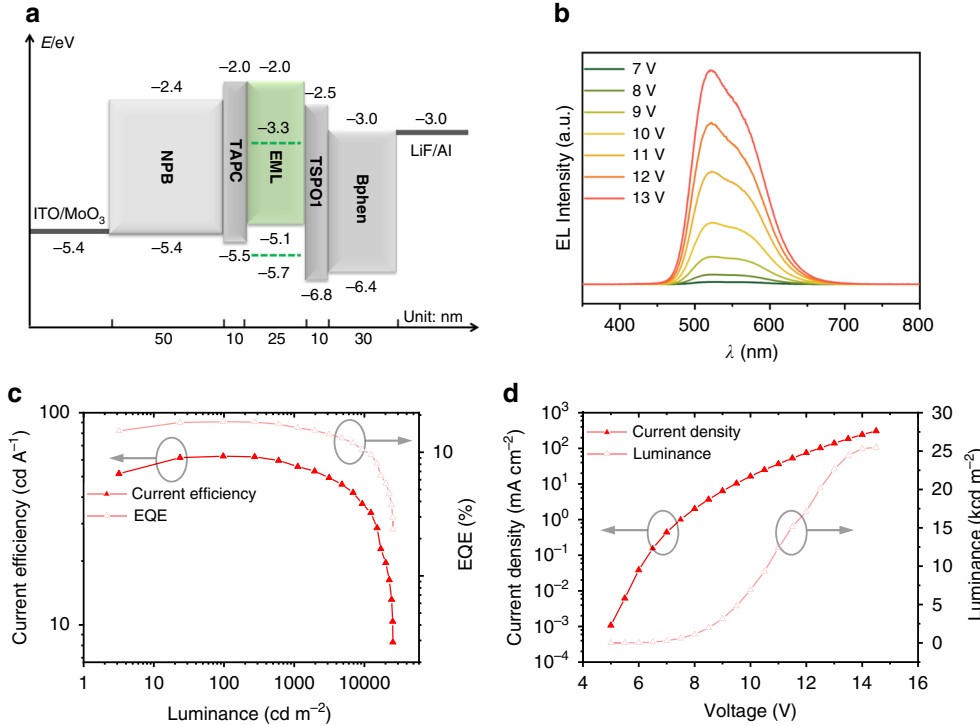

**Fig. 4 The device structure and electroluminescence properties of Eu$X_2$-N$_8$ compounds. a** The optimized OLEDs device structure with frontier orbital energy levels of all organic materials and corresponding thickness. The HOMO and LUMO level of EuI$_2$–N$_8$ are noted as green dash lines in the EML. **b** Electroluminescence spectra of the champion device at varying voltage from 7 V to 13 V. **c** Current efficiency–luminance–external quantum efficiency (CE–L–EQE) curve of the champion device. **d** Current density–voltage–luminance (J–V–L) curve of the champion device.

6.5 V, 25,470 cd m$^{-2}$, 62.4 cd A$^{-1}$, and 17.7%, respectively. These results shown in Fig. 4 far exceed the only previously reported Eu$^{2+}$-based OLEDs, with EQE$_{max}$ of 0.01%, $L_{max}$ of 10 cd m$^{-2}$ and V$_{on}$ of 20 V[26]. In addition, the electroluminescence lifetimes of the champion device are determined at 2.5 mA cm$^{-2}$ and 10 mA cm$^{-2}$, which are little shorter than tris(2-phenylpyridine) iridium (Ir(ppy)$_3$) based control device, a well-studied phosphorescence emitter in the same spectral region (Supplementary Fig. 18). It should be noted that both devices showed very short lifetime, since which is not only related to the emission material, but also host material, charge transport material, device fabrication, seal conditions, and so on[37].

It is notable that the Eu$X_2$–N$_8$ devices have rather high V$_{on}$ values over 6 V considering the bandgap of host material is only 3 eV (~400 nm). To understand this phenomenon, the photophysical properties of films fabricated by doping 10 wt% Eu$X_2$–N$_8$ in m-MTDATA onto quartz substrates in a vacuum chamber at high vacuum (10$^{-5}$ Pa) were studied. The pure films of Eu$X_2$–N$_8$ were also fabricated as reference. The emission of doping films is mainly located as two bands, where the 400–470 nm band ($\tau$~1.2 ns) is attributed as fluorescence from host materials and the 500–650 nm band ($\tau$~10$^2$ ns) is from Eu$X_2$–N$_8$ in Supplementary Fig. 19. The excitation spectra show that in pure film, the two main excitation bands are located at around 320 nm and 390 nm, which is close to the studies showed in solution and solid state. The two doping films have almost identical excitation band at 350 nm, indicating the photoenergy first excites host materials and then transfers to doping Eu$^{2+}$ complexes, without complete energy transfer in the photoluminescence process. Intriguingly, the electroluminescence spectra only exhibit emission from Eu$^{2+}$ complexes with doping concentrations (7 wt%) lower than those in the photoluminescence study. Thus, we tentatively propose that the carrier recombination dominantly occurs in the doping materials instead of host materials, where the excitation of ligand results in a high V$_{on}$.

## Discussion

In summary, four Eu$^{2+}$-containing azacryptates Eu$X_2$–N$_n$ ($X$ = Br, I, $n$ = 4, 8) were synthesized, showing promising photoluminescent properties: high PLQY (~100% for N$_8$ complexes), short excited-state lifetime (10$^2$ ns) and easily tunable emission by ligand field. Intriguingly, EuI$_2$–N$_8$ exhibits reversible mechanochromic property under grinding, which is attributed to the potential flexibility of N$_8$ ligand and recrystallization. Furthermore, the Eu$X_2$–N$_8$ complexes were chosen as the emissive materials in OLEDs due to their good air-/thermal-stability. After optimization of design, the best device showed excellent performance with a maximum EQE of 17.7% and luminance of 25,470 cd m$^{-2}$. Our work deepens the understanding of photoluminescence and electroluminescence properties in Eu$^{2+}$ complexes and proves their promising applications in OLEDs.

## Methods

All chemical reagents used in the synthesis process were commercially available and were used as received unless otherwise mentioned. The N$_4$ ligand was commercially available. $^1$H-NMR spectra were recorded on a Bruker-400 MHz NMR spectrometer. Tetramethylsilane (TMS) was used as an internal reference for the chemical shift correction, where δ(TMS) equals 0. Elemental analyses were performed on a VARIO EL analyzer (GmbH, Hanau, Germany). All the synthesis of Eu$^{2+}$ complexes was conducted in glovebox. All spectral tests of solid Eu$^{2+}$ complexes were carried out by paraffin encapsulation between two quartz plates and the solution was protected by capped cuvettes under N$_2$ atmosphere. The commercially available paraffin was purified by oxidation using KMnO$_4$ and column chromatography to remove fluorescent whitening agents.

**Synthesis**. *1,4,7,10,13,16,21,24-octaazabicyclo[8.8.8]hexacosane (N$_8$ ligand)*: The synthesis of N$_8$ ligand is carried out by an improved version of a reported method[38]. *Tris*(2-aminoethyl)amine (4.9 g, 33.5 mmol), NEt$_3$ (12 mL), and 2-propanol (250 mL) were added to a 2-neck 1-L round-bottom flask equipped with mechanical stirring and a drip funnel containing a dilute solution of glyoxal (7.5 g). The flask was cooled to −78 °C and the glyoxal solution was added slowly (1 drop s$^{-1}$). After the completion of addition, the yellow solution was stirred at room temperature overnight. Then the solvent was removed under vacuum at 40 °C, yielding a yellow solid which was dispersed in 300 mL CHCl$_3$ and stirred for 2 h with the generation of lots of yellow

translucent gels. The gels were removed by filtration and the resulting $CHCl_3$ was removed under vacuum at 40 °C. The crude intermediate was dissolved in 300 mL MeOH, cooled with ice water. Excess $NaBH_4$ (14 g) was gradually added to the solution to prevent an intensive reaction. The cloudy solution was stirred for 4 h and the solvent was removed under vacuum yielding white solid, which was extracted by $CH_2Cl_2$ (200 mL×3). The removal of $CH_2Cl_2$ gave the crude product $N_8$. Further purification was conducted by thermal gradient sublimation (160−80 °C) at low pressure (~5 Pa). $^1H$-NMR (400 MHz, $D_2O$): δ 2.79 (s, 12H), 2.75 (m, 12H), 2.58 (m, 12H).

*EuBr$_2$-N$_4$*: $EuBr_2$ (78 mg, 0.29 mmol) was dissolved in 6 mL MeOH in a clean glass bottle. $N_4$ (85 mg, 0.50 mmol) was dissolved in 3.5 mL MeOH, which was slowly added to the $EuBr_2$ solution without stirring. The colorless solution turned orange-red and red crystals suitable for SCXRD analysis formed as the evaporation of solvent (yield is 85% based on Eu). Elemental analysis for $C_{16}H_{40}Br_2EuN_8$, C, 29.28%, H, 6.14%, N, 17.07%. Found: C, 29.05%, H, 6.05%, N, 16.69%.

*EuI$_2$-N$_4$*: $EuI_2$ (55 mg, 0.14 mmol) was dissolved in 6 mL MeOH in a clean glass bottle. $N_4$ (50 mg, 0.29 mmol) was dissolved in 3.5 mL MeOH, which was slowly added to the $EuI_2$ solution without stirring. The light-yellow solution turned orange-red and red crystals suitable for SCXRD analysis formed soon after the completion of mixing (yield is 70% based on Eu). Elemental analysis for $C_{16}H_{40}EuI_2N_8$, C, 25.61%, H, 5.37%, N, 14.93%. Found: C, 25.84%, H, 5.54%, N, 14.42%.

*EuBr$_2$-N$_8$*: $EuBr_2$ (0.156 g, 0.500 mmol) was dissolved in 10 mL MeOH in a 50-mL round-bottom flask under magnetic stirring. $N_8$ (0.185 g, 0.500 mmol) was dissolved in 10 mL MeOH, which was slowly added to the $EuBr_2$ solution. The colorless solution turned orange. The solvent was removed under reduced pressure to get crude product. (Yield is 86% based on Eu). The complex was further purified by thermal gradient sublimation (320–250–60 °C) at low pressure ($10^{-5}$ Pa). Green crystals suitable for SCXRD analysis were obtained by slow evaporation of MeOH as a solvent. Elemental analysis for $C_{18}H_{42}Br_2EuN_8$, C, 31.68%, H, 6.20%, N, 16.42%. Found: C, 31.75%, H, 6.18%, N, 16.39%.

*EuI$_2$-N$_8$*: $EuI_2$ (0.260 g, 0.64 mmol) was dissolved in 10 mL MeOH in a 50-mL round-bottom flask under magnetic stirring. $N_8$ (0.260 g, 0.70 mmol) was dissolved in 10 mL MeOH, which was slowly added to the $EuI_2$ solution. The light-yellow solution turned orange and green crystals suitable for SCXRD analysis formed as the evaporation of solvent without stirring (yield is 72% based on Eu). Elemental analysis for $C_{18}H_{42}I_2EuN_8$, C, 27.85%, H, 5.45%, N, 14.43%. Found: C, 28.04%, H, 5.41%, N, 14.28%.

## Data availability

All the data in manuscript and supporting information are available from the corresponding author upon reasonable request.

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

## Acknowledgements

This work was supported by the National Key R&D Program of China (2016YFB0401001, 2017YFA0205100), the Beijing Natural Science Foundation (2202015), and High-performance Computing Platform of Peking University.

## Author contributions

Z.L. proposed and designed this project. J.L., L.W., and Z.Z. conducted most of the experiments. B.S. and H.L. helped in synthesizing ligands. G.Z. measured the PLQY values. Z.L., Z.B., and J.L. discussed the results and wrote the manuscript.

## Competing interests

The authors declare no competing interests.
