## [Peer Review File · Nature Communications]

Reviewers' Comments:

Reviewer #1:

Remarks to the Author:

1. In the introduction part, the authors think the Eu-based complex are strongly limited by their air stability. What cause the low air stability of the reported Eu-based complexes? The authors propose that the steric effect of cryptate ligands and coordinate interaction can improve the stability of Eu-based complex. Why? The design strategy of these complexes should be highlighted.
2. In Page 4, the authors claim OLEDs based on EuI1-N8 show the excellent OLED performance of 17.7%. It is very impressive for the Eu-based complexes as a breakthrough. However, I do not agree the performance can be comparable to the most state-of-the-art OLEDs with phosphorescent emitters and TADF emitters.
3. In Figure 4 c and d, the arrows should be added to indicate the coordinate axis of the corresponding curves.
4. The HOMO and LUMO of these Eu-based complexes should be investigated, which are very important for the design of OLEDs.
5. All the OLED devices show high turn-on voltage of 6V. The authors think this can be attributed to the bandgap of host materials of about 3 eV. I suggest the authors to further optimize the device structure by host choice and device optimization.

Reviewer #2:

Remarks to the Author:

In the present manuscript NCOMMS-20-16607-T, the authors discuss new Eu(II)-coordinated compounds with very good photoluminescence and stability properties. These materials have been used in organic light-emitting diodes (OLEDs) within this study, where high efficiency values have been reported. Overall, the results presented are very interesting and encouraging for this emitter material class for novel applications. The presentation is detailed and sound. There are a couple of points that I would suggest to address prior before a final recommendation. Please find these below:

1. For the compounds, the transient photoluminescence (PL) decays have been measured and analyzed. Some of the decays are not mono exponential. Therefore it would be important to explain and state the analysis that has been used to derive the lifetime values.
2. I am not an expert in analytics in chemistry. The figure 3c and d show some buried volume ... I do not know this measure and could not easily understand it with the text provided. I would encourage to revise the manuscript to explain this analysis properly so that non-experts to understand it.
3. If I am not mistaken, there is no reference to Figure 4 in the main text. Please check carefully.
4. The authors discuss the stability of the emitters in PL with very good detail. What researchers working in the field of OLEDs would appreciate much, would be the inclusion of electroluminescence lifetime data. This is very important to rank in this finding to other emitter classes. I do know of course that this is likely a very early study, where the values might not be optimized, but still, it would help. I would actually suggest the following. One respective measurement LT50 or LT70, depending, how good the devices are, at a fixed current density in the range of 10-15 mA/cm². And one control device that uses the same OLED stack with the only change to use an established emitter in the same spectral region (can either be Ir(ppy)₃ as phosphorescent or 4CzIPN as TADF emitter. The lifetime measurement should be done at the same current density.

5. I am not certain, the representation of Figure 4a makes sense. The y-axis represents the maximum luminance, but I am wondering, how this maximum luminance is determined. Is it the value before failure, or at a given endpoint of the jV sweep. If the latter, than this value is absolutely arbitrary, as a higher value could have been obtained by sweeping the device to higher current densities. Sill, L and EQE are somewhat correlated anyway, why this presentation does not offer new insight. Something like EQE vs. PL decay lifetime, or EQE vs. OLED EL lifetime (stability) would be interesting visualizations. I would strongly encourage the authors to revise this...

Referee #1:

Question 1: In the introduction part, the authors think the Eu-based complex are strongly limited by their air stability. What cause the low air stability of the reported Eu-based complexes? The authors propose that the steric effect of cryptate ligands and coordinate interaction can improve the stability of Eu-based complex. Why? The design strategy of these complexes should be highlighted.

Reply: Thank you very much for your suggestions. According to the standard potential: $\varphi(\text{Eu}^{3+}/\text{Eu}^{2+}) = -0.38 \text{ V}$, the Eu^{2+} ions are expected to be highly unstable towards O_2 . Although it is notable that Eu^{2+} -doping inorganic solids are generally air-stable, this is attributed to the rigid structure of host matrix, preventing O_2 from oxidation, as well as the low concentration of Eu^{2+} . In the field of Eu^{2+} -based complexes, most of the work didn't mention the air stability, where the synthesis and characterization were conducted in gloveboxes or under N_2 protection. The N_8 and N_4 ligands provide strong steric effect and high coordination number. The steric effect has been explored to stabilize highly active metal, e.g. Rh(I) , In(I) , Au(I) , Zn(0) .^{1,2,3} On the other hand, Eu^{2+} ion is larger in radius comparing with Eu^{3+} and expected to have a saturated coordination number of 8-12. Improving the coordination interaction between the ligand and Eu^{2+} can largely enhance the thermodynamic stability.

Accordingly, we have added "Despite of the advantages mentioned, Eu^{2+} complexes are strongly limited by their poor air stability according to the standard potential $\varphi(\text{Eu}^{3+}/\text{Eu}^{2+}) = -0.38 \text{ V}$." and "We propose that the steric effect of cryptate ligands and coordinate interaction could improve the stability of Eu^{2+} complexes. Steric effect prevents Eu^{2+} from O_2 by more rigid structure. Improving the coordination interaction between the ligand and Eu^{2+} can largely enhance the thermodynamic stability." in the revised manuscript to explain the low air stability of the reported Eu-based complexes, and to highlight the design strategy of our complexes, respectively.

Question 2: In Page 4, the authors claim OLEDs based on $\text{EuI}_2\text{-N}_8$ show the excellent OLED performance of 17.7%. It is very impressive for the Eu-based complexes as a breakthrough. However, I do not agree the performance can be comparable to the most state-of-the-art OLEDs with phosphorescent emitters and TADF emitters.

Reply: Thanks for your correction. We have deleted this statement in the sections of electroluminescence performance and conclusion.

Question 3: In Figure 4 c and d, the arrows should be added to indicate the coordinate axis of the corresponding curves.

Reply: Thanks for your suggestion. We have revised the Fig. 4(c) and 4(d) in the manuscript. Please see the revised figure below.

Figure 1. The revised Fig. 4(c) and 4(d) in manuscript.

Question 4: The HOMO and LUMO of these Eu-based complexes should be investigated, which are very important for the design of OLEDs.

Reply: Thanks for your suggestion. We measured the HOMO in the two N₈ complexes by ultraviolet photoelectron spectroscopy (UPS). Based on the data deduced from the ultraviolet photoelectron spectra and the edge of ultraviolet absorption spectra ($\lambda = 527$ nm), the HOMO and LUMO energy levels of the two complexes are calculated as follows:

EuBr₂-N₈:

$$E_{\text{HOMO}} = -(h\nu - E_{\text{B}} + E_{\text{A}}) = -(21.22 \text{ eV} - 18.86 \text{ eV} + 3.19 \text{ eV}) \approx -5.6 \text{ eV}$$

$$E_{\text{LUMO}} = E_{\text{HOMO}} + hc/\lambda = -5.6 \text{ eV} + 1240/527 \text{ eV} \approx -3.2 \text{ eV}$$

EuI₂-N₈:

$$E_{\text{HOMO}} = -(h\nu - E_{\text{B}} + E_{\text{A}}) = -(21.22 \text{ eV} - 17.70 \text{ eV} + 2.15 \text{ eV}) \approx -5.7 \text{ eV}$$

$$E_{\text{LUMO}} = E_{\text{HOMO}} + hc/\lambda = -5.7 \text{ eV} + 1240/527 \text{ eV} \approx -3.3 \text{ eV}$$

The related data and discussion have been added in the revised manuscript and SI.

Figure 2. The ultraviolet photoelectron spectroscopy of (a) EuBr₂-N₈ and (b) EuI₂-N₈.

Question 5: All the OLED devices show high turn-on voltage of 6V. The authors think this can be attributed to the bandgap of host materials of about 3 eV. I suggest the authors to further optimize the device structure by host choice and device optimization.

Reply: Thanks for the suggestion. The optimized devices exhibit higher turn-on voltage (~6 V) comparing with the state-of-art OLED devices. Traditionally, in the process of electroluminescence, exciton combination happens in the host materials, followed by highly efficient energy transfer from host to dopant. Thus, the theoretical $V_{on, min}$ should be close to the bandgap of host materials. However, the turn-on voltage of 6 V is much higher than the bandgap of m-MTDATA (3 eV, ~400 nm), the host material used in our devices.

Then in our manuscript we compared the PL spectra with EL spectra of $\text{EuX}_2\text{-N}_8$ complexes. Surprisingly, the 10 wt% doping films show obvious emission from host material while the 7 wt% EL spectra show pure emission from complexes. This indicates the exciton recombination might occur directly in the doping materials (Eu-complexes). We tentatively attribute the high V_{on} to the excitation of ligand, not the host materials.

As for your second point, we have tested various host materials in the very first step and fabricated doping films at a fixed doping concentration of $\text{EuBr}_2\text{-N}_8$. These films, however, are highly sensitive to air as expected. Hence, it would be impossible for us to measure the accurate PLQY of each film. Showing below are the pictures of doping films with different host materials under excitation of 365 nm in glovebox. Although PLQY values are not accessible, it's obvious that TAPC (film 6) and m-MTDATA (film 9) exhibit the best performances by bare eyes. It is worth noting that the luminescent intensity is not proportional to the actual PLQY because light absorbance may be different. We have attached this part of work in section 1, SI.

Figure 3. The $\text{EuBr}_2\text{-N}_8$ pure film and $\text{EuBr}_2\text{-N}_8$ doping films using different host materials. Pictures were taken under UV excitation inside the glovebox. **1:** TCTA; **2:** CBP; **3:** TPBi; **4:** DPEPO; **5:** NPB; **6:** TAPC; **7:** $\text{EuBr}_2\text{-N}_8$ pure film; **8:** DIC-TRZ; **9:** m-MTDATA.

Referee #2:

Comments: In the present manuscript NCOMMS-20-16607-T, the authors discuss new Eu(II)-coordinated compounds with very good photoluminescence and stability properties. These materials have been used in organic light-emitting diodes (OLEDs) within this study, where high efficiency values have been reported. Overall, the results presented are very interesting and encouraging for this emitter material class for novel applications. The presentation is detailed and sound. There are a couple of points that I would suggest addressing prior before a final recommendation. Please find these below:

Question 1: For the compounds, the transient photoluminescence (PL) decays have been measured and analyzed. Some of the decays are not mono exponential. Therefore, it would be important to explain and state the analysis that has been used to derive the lifetime values.

Reply: Thanks for your advice. We have updated the details of data fitting in the SI. If I didn't misunderstand your question, you referred to the PL decays in Fig. 2(b), (d) and (e). In our view, the transient decay shows good mono-exponential feature. Taking the transient spectra of the four complexes (solid) as example, the black dash line is the fitting line ($y = kx + b$) and the fitting range is from 0 to 5000 ns. The data after 5000 ns is not included because of low signal-to-noise ratio. As you can see below, the R^2 is close to 1 for each complex, indicating valid fit to the decay lifetime.

Figure 4. Linear fit of the PL decay data.

The data is processed using OriginPro 2020b (academic version). The datapoints were shifted to make sure that t_0 corresponds to the maximum intensity (I_0). The y axis is using natural logarithmic coordinate. For an ideal decay process without other competing path, the luminescent intensity will undergo exponential decay. Hence, we can determine the decay lifetime τ as the reciprocal of the slope.

$$I(t) = I_0 e^{-\frac{t}{\tau}}$$

$$\ln[I(t)] = \text{Const.} - \tau^{-1}t$$

Question 2: I am not an expert in analytics in chemistry. The figure 3c and d show some buried volume ... I do not know this measure and could not easily understand it with the text provided. I would encourage to revise the manuscript to explain this analysis properly so that non-experts to understand it.

Reply: Thanks for your suggestion. We apologize for insufficient explanation of V_{bur} plots. The V_{bur} calculation is to visualize the coordination environment around the metal and compare the steric effect of ligands. The calculations were conducted in the web application named SambVca (<https://www.molnac.unisa.it/OMtools/sambvca2.1/index.html>).

(1) Brief introduction: The two key outputs are $\%V_{\text{bur}}$ and the topographic steric map. $\%V_{\text{bur}}$ is defined as the fraction of volume of ligand over the total volume of sphere centered on the metal. It determines the steric effect of a given ligand regard to the first coordination sphere. The topographic steric maps are the chemical analog to physical maps for geographical features. The color change from blue to red represents the ligand position with reference to the metal in z axis.

(2) Details of calculation for the Eu(II) complexes: In order to get a view inside the coordination sphere, the top of the N_8 ligand and the Eu^{2+} are omitted in the calculation as illustrated from Fig. 5(a) to 5(b). The coordinate system is established, where Eu^{2+} being the coordinate origin. Fig. 5(c) is the top view of the complex.

Figure 5. The illustration of $\%V_{\text{bur}}$ calculation: (a) the crystal structure of $\text{EuBr}_2\text{-N}_8$; (b) eliminating the top of the N_8 ligand and Eu^{2+} center, establishing the coordinate system; (c) top view of (b).

We have added a brief explanation both in manuscript and the SI. The details of calculation method and parameters are included in SI.

Question 3: If I am not mistaken, there is no reference to Figure 4 in the main text. Please check carefully.

Reply: Thanks for your correction. We have referred Figure 4 in the revised manuscript.

Question 4: The authors discuss the stability of the emitters in PL with very good detail. What researchers working in the field of OLEDs would appreciate much, would be the inclusion of electroluminescence lifetime data. This is very important to rank in this finding to other emitter classes. I do know of course that this is likely a very early study, where the values might not be optimized, but still, it would help. I would actually suggest the following. One respective measurement LT50 or LT70, depending, how good the devices are, at a fixed current density in the range of 10-15 mA/cm². And one control device that uses the same OLED stack with the only change to use an established emitter in the same spectral region (can either be Ir(ppy)₃ as phosphorescent or 4CzIPN as TADF emitter. The lifetime measurement should be done at the same current density.

Reply: Thanks for your suggestion. Indeed, the device lifetime would be of high significance for further research and comparing with other luminescent materials. We chose Ir(ppy)₃ as the control group. The figures shown below is the EL lifetime with fixed current density of (a) 10 mA/cm² (b) 2.5 mA/cm². The LT50 values are noted in the figures. The related data and discussion have been added in the revised manuscript and SI.

Figure 6. The EL lifetime with fixed current density of (a) 2.5 mA/cm² (b) 10 mA/cm². The LT50 values are noted in the figures. The controlled device used Ir(ppy)₃ as emitters.

Question 5: I am not certain, the representation of Figure 4a makes sense. The y-axis represents the maximum luminance, but I am wondering, how this maximum luminance is determined. Is it the value before failure, or at a given endpoint of the j-V sweep? If the latter, then this value is absolutely arbitrary, as a higher value could have been obtained by sweeping the device to

higher current densities. Still, L and EQE are somewhat correlated anyway, why this presentation does not offer new insight. Something like EQE vs. PL decay lifetime, or EQE vs. OLED EL lifetime (stability) would be interesting visualizations. I would strongly encourage the authors to revise this...

Reply: Thanks for your suggestion. We have clarified the maximum luminance as the max value before failure (starting to drop). Hence, when comparing with different devices, the two parameters, L_{\max} and EQE_{\max} , are not strictly correlated. Some devices might have similar EQE_{\max} while differ a lot in L_{\max} . Our purpose of Fig. 4(a) is to show how to balance these two and choose the optimized condition in each section. For example, if comparing the device A and device B with the coordinate (EQE_{\max} , L_{\max}) of (11.8, 6652) and (12.4, 4059), respectively, we chose the device structure of A for further optimization.

As for your second point, we are not quite sure if we correctly understand your advice. The PL decay lifetime should be very similar in these devices because of the same doping concentrations in the emissive layer. The EQE_{\max} vs. EL lifetime could be an interesting point for further study. However, we think the lifetime of the champion device would be enough for now and it would be very time-consuming and meaningless if we reproduce all the devices only to determine the decay lifetime, since these devices are not optimized. In all, we believe the comprehensive study on EL lifetime vs. different device structure should be focused on when the research on Eu^{2+} complexes in OLEDs is much more mature and solid.

Reference:

- (1) Tolman, C. A. Phosphorus Ligand Exchange Equilibriums on Zerovalent Nickel. Dominant Role for Steric Effects. *J. Am. Chem. Soc.* **1970**, *92* (10), 2956–2965. <https://doi.org/10.1021/ja00713a007>.
- (2) Kato, T.; Takahashi, S.; Nakaya, K.; Baceiredo, A.; Saffon-Merceron, N.; Massou, S.; Nakata, N.; Hashizume, D.; Branchadell, V.; Pastor, M. F. Synthesis of A Stable N-Hetero-Rh(I)-Metallacyclic Silanone. *Angew. Chem. Int. Ed.* *n/a* (n/a). <https://doi.org/10.1002/anie.202006088>.
- (3) Hill, M. S.; Hitchcock, P. B.; Pongtavornpinyo, R. Neutral Carbene Analogues of the Heaviest Group 13 Elements: Consideration of Electronic and Steric Effects on Structure and Stability. *Dalton Trans.* **2005**, No. 2, 273–277. <https://doi.org/10.1039/B414462G>.

Reviewers' Comments:

Reviewer #1:

Remarks to the Author:

The authors have revised the manuscript properly. Now I am in favor of publication of the manuscript.

Reviewer #2:

Remarks to the Author:

The current manuscript NCOMMS-20-16607A is a first revision of an original submission at Nature Communications. The manuscript has received two reviewers reports and this current revision contains all the changes and additions based on the reviews. Overall, the authors have addressed all points I raised during the original review, most of them to complete satisfaction. Below are some points which I think are worth to address in a minor revision. With this future improvements, I am certain that the manuscript can be accepted for publication.

1. Referring to the discussion of the PL decay fits. I agree that most of the decays are mono exponential (Figure 4 in the rebuttal letter, but some are not. In particular, the fits to data of EuBr₂-N8 and EuI₂-N8 are not well fitted with one exponent. This would become obvious, if the authors would plot the fit residuals. For EuBr₂-N8, the fit misses the initial normalized value of 1 by almost 50%. A more careful analysis would be helpful.

2. During the revision, the authors have studied the PL of the EuBr₂-N8 in different host materials - the result is now shown in the Section 1 of the SI. While I really like this effort, I think the photographs alone do not satisfy the case and reader. While I agree that PLQY values are maybe hard to obtain for all the combinations, the face PL emission spectrum (normal to the substrate) should be easy to acquire. This data is of key importance for the reader to judge and evaluate the relative suitability of the different potential host materials. Without it, it is not possible to distinguish between single, double band emission, possible shoulders in the spectra - all of which would indicate the relative working principle of the combination under study.

3. I am still not convinced that the L_{max} vs. EQE_{max} plot is meaningful. I might try to explain again. If the L_{max} is determined at the point, where the device is about to break down, then this value is nothing more than an unphysical R&D value that is used to identify a best working system within the variations (in material and thicknesses etc.) the authors carried out (actually this is what they did). My criticism goes to the direction that it is not a value that can be generalized or related back to the emitter properties. I acknowledge that this method helped to identify suitable device optimization ranges, but unfortunately it does not tell more.

Reviewer #1:

Comments: The authors have revised the manuscript properly. Now I am in favor of publication of the manuscript.

Reply: Thanks for your advice and time.

Reviewer #2:

Comments: The current manuscript NCOMMS-20-16607A is a first revision of an original submission at Nature Communications. The manuscript has received two reviewers reports and this current revision contains all the changes and additions based on the reviews. Overall, the authors have addressed all points I raised during the original review, most of them to complete satisfaction. Below are some points which I think are worth to address in a minor revision. With this future improvements, I am certain that the manuscript can be accepted for publication.

Question 1: Referring to the discussion of the PL decay fits. I agree that most of the decays are mono exponential (Figure 4 in the rebuttal letter, but some are not. In particular, the fits to data of $\text{EuBr}_2\text{-N}_8$ and $\text{EuI}_2\text{-N}_8$ are not well fitted with one exponent. This would become obvious, if the authors would plot the fit residuals. For $\text{EuBr}_2\text{-N}_8$, the fit misses the initial normalized value of 1 by almost 50%. A more careful analysis would be helpful.

Reply: Thanks for your corrections. We have carefully retested the transient spectra of $\text{EuBr}_2\text{-N}_8$ and $\text{EuI}_2\text{-N}_8$. The new spectrum of $\text{EuBr}_2\text{-N}_8$ exhibits a different value of decay (997 ns) comparing with our previous data and shows better mono-exponential feature. The decay lifetime of $\text{EuI}_2\text{-N}_8$ (867 ns) is almost the same as our previous data (864 ns). Hence, the updated transient spectra are better fitted by mono-exponential decay. We have updated the new decay lifetime of $\text{EuBr}_2\text{-N}_8$ in manuscript (Fig. 2 (b) and Table 2).

Question 2: During the revision, the authors have studied the PL of the $\text{EuBr}_2\text{-N}_8$ in different host materials - the result is now shown in the Section 1 of the SI. While I really like this effort, I think the photographs alone do not satisfy the case and reader. While I agree that PLQY values are maybe hard to obtain for all the combinations, the face PL emission spectrum (normal to the substrate) should be easy to acquire. This data is of key importance for the reader to judge and evaluate the relative suitability of the different potential host materials. Without it, it is not possible to distinguish between single, double band emission, possible shoulders in the spectra - all of which would indicate the relative working principle of the combination under study.

Reply: Thanks for your advice. We have tested the photoluminescence spectra of selected films. In the picture shown below, film 1 (TCTA), film 6 (TAPC), film 8 (DIC-TRZ) and film 9 (m-MTDATA) were measured. The other films are too weakly emissive to get a valid data (as you can see the signal-to-noise ratios of film 1 and 8 are already low).

(a)

(b)

Question 3. I am still not convinced that the L_{\max} vs. EQE_{\max} plot is meaningful. I might try to explain again. If the L_{\max} is determined at the point, where the device is about to break down, then this value is nothing more than an unphysical R&D value that is used to identify a best working system within the variations (in material and thicknesses etc.) the authors carried out (actually this is what they did). My criticism goes to the direction that it is not a value that can be generalized or related back to the emitter properties. I acknowledge that this method helped to identify suitable device optimization ranges, but unfortunately it does not tell more.

Reply: Thanks for your explanation. We have replaced this plot with the device structure as shown below.

Reviewers' Comments:

Reviewer #2:

Remarks to the Author:

In this section revision, the authors have addressed the additional comments I made in the second round of review. All of their changes and additions are fully satisfactory and will help to improve the overall quality of the report. Thanks for this effort, I am in full support of this publication.

Reviewer #2:

Comments: In this section revision, the authors have addressed the additional comments I made in the second round of review. All of their changes and additions are fully satisfactory and will help to improve the overall quality of the report. Thanks for this effort, I am in full support of this publication.

Reply: We sincerely thank your advice and insight.